# Identifying daily-living features related to loneliness: A causal machine learning approach

Yuning Wang[1]*, Jennifer Auxier[2], Mark Amayag[3], Iman Azimi[4], Amir M. Rahmani[4], Pasi Liljeberg[1], Anna Axelin[3]

1 University of Turku, Department of Computing, Turku, Finland, 2 University of British Columbia, Vancouver, British Columbia, Canada, 3 University of Turku, Department of Nursing Science, Turku, Finland, 4 University of California, Department of Computer Science, Irvine, California, United States of America

* yuning.y.wang@utu.fi

## Abstract

### Background

Loneliness is a distressing feeling that influences well-being. Immigrants' experience of acculturation to a new dominant culture places them at risk for maladaptive behaviors and daily rhythms leading to loneliness. Identifying daily-living features that causally influence loneliness is essential for developing effective preventive mental health screening.

### Objective

To identify the important daily living-features related to loneliness for the development of robust screening solutions using causal machine learning for health providers working with first-generation immigrants.

### Methods

We monitored 39 immigrants in Finland for 28 days using mobile devices and wearables under free-living conditions. Data included ecological momentary assessments of loneliness, social interactions, physical activity, sleep, and cardiac features. We estimated the average treatment effect (ATE) of each daily-living feature (treatment variable) on loneliness scores (outcome) and validated the robustness of causal estimates using three refutation techniques.

### Results

Our results reveal the ATE of various daily-living features on loneliness. Features such as longer outgoing call durations (ATE = 0.197, $p < 0.001$), higher LF/HF ratio (ATE = 0.129, $p < 0.0001$), higher respiratory rate (ATE = 0.144, $p < 0.001$), and increased inactivity (ATE = 0.130, $p < 0.001$) causally increased loneliness.

**Data availability statement:** All data used in this study have been fully de-identified to protect participant privacy. The de-identified dataset is publicly available on the Dryad Digital Repository at: https://doi.org/10.5061/dryad.qz612jmrn.

**Funding:** The author(s) received no specific funding for this work.

**Competing interests:** The authors have declared that no competing interests exist.

Conversely, certain features exhibit negative ATEs, such as higher activity calories (ATE = −0.174, p < 0.001), sleep RMSSD (ATE = −0.128, p < 0.001), longer home duration (ATE = −0.107, p < 0.001), and more sleep time (ATE = −0.103, p < 0.001) mitigated loneliness.

## Conclusions

Daily-living features, including social interactions, activity, sleep, and cardiac features, causally influence loneliness. Our findings provide a basis for loneliness screening targeting immigrant populations. Future work should refine the measurement and incorporate contextual information to establish more reliable causal links in real life.

## Introduction

Immigrants' experiences of adapting to a new dominant culture are associated with risk factors for social isolation, loneliness and suicide [1] Many elements of daily life place undue stress, and emotional discomfort on immigrants, such as, searching for and maintaining a place of residence, and employment/schooling for themselves and/or their family members [2]. The daily experience of loneliness has been shown to impact health patterns such as sleep quality, cardiovascular health, cognition, emotion, and behavior [3–5]. Maladaptive health patterns, such as poor sleep quality, sedentary lifestyle, and the lack of social support have been linked to depression, stress and feelings of loneliness [6,7]. Compounding these risk factors, lifestyle conditions and habits associated with experiencing immigration to a new country place individuals at risk of severance and loss of meaningful social connections.

From the perspective of health care providers, it can be challenging to detect risk factors for loneliness and to support evidenced based prevention strategies for social isolation because of the complexity of loneliness. Loneliness can be perceived, or real, and an experience that is transient for a person or might persist in becoming a chronic state [8,9]. Recently, there has been momentum in the development and use of real time and remote monitoring practices around the world [5,10]. Methods such as, Ecological Momentary Assessment (EMA) show promise for improving monitoring of maladaptive health patterns that put immigrants at risk of loneliness, however screening for the presence of daily-living patterns linked to risk of loneliness has not been developed [5].

Further, loneliness prevention and management could improve a wide range of health outcomes [3–5]. Interventions for loneliness up to this date are traditional, provided at most when lonely individuals experience worsening symptoms or when they decide to seek medical assistance. Moreover, many of these interventions in their research phase exclude technology use, often utilizing unvalidated, or internally designed tools [11–13]. Real-time support and prevention for loneliness could be developed through the early detection of loneliness risk factors.

Recent advancements in technology have allowed researchers to move beyond traditional self-reported measures and use real-time monitoring of daily behaviors to

observe their relationship with mental health outcomes [10,14]. Wearable devices like smartwatches can capture physiological markers such as heart rate variability (HRV), physical activity, sleep quality, and skin conductance [15]. These signals are closely related to emotional states, including stress and loneliness. For instance, researchers identified the participant's stress level by monitoring their body temperature, sweat, and motion rate during physical activity [16]. In addition to wearables, smartphone applications are frequently used to track behavioral and social interaction patterns such as location data, screen time, and communication patterns. For example, less number of phone calls indicates increasing social anxiety [17].

Traditionally, research analyzing the relationship between daily-living features and loneliness has relied heavily on conventional statistical analysis. For example, Johnson and colleagues found subjective sleep quality can be significantly predicted by daily average loneliness using hierarchical linear models [5]. In recent years, machine learning methods have gained traction in this field, offering a more data-driven approach to identifying important predictors of loneliness. For instance, Jafarlou and colleagues leveraged Random Forests algorithm and revealed that the number of notifications in the "lifestyle" category is the most predictive feature for college students' loneliness [18]. However, these approaches often still focus on predictive power rather than addressing the underlying causality. This means that while machine learning can identify the most relevant features, it cannot determine whether changes in these behaviors cause changes in loneliness.

In response to this limitation, causal inference methods are becoming increasingly important.

Causal machine learning (CML) approaches provide a framework for not only identifying important predictors of loneliness but also establishing the directionality and strength of causal relationships. For example, double machine learning (DML) models are commonly used to estimate the conditional average treatment effect (CATE) by controlling for confounding factors [19]. Additionally, tools such as CausalML [20] and DoWhy [21] provide frameworks to model the causal impact of treatment on outcomes.

In this study, we aimed to test and implement methods for monitoring observational behavioral and physiological data of immigrants in Finland to understand which daily-living features have causal links to perceptions of loneliness using a CML analysis method and to guide the development of a screening tool for risk factors of loneliness.

## Materials and methods

### Study design

Prospective observational longitudinal study with 28 days follow-up was conducted between September 2022 and June 2023.

### Participants and recruitment

Adult immigrants aged 18–65 were recruited to participate in the study. Eligibility criteria required participants to: 1) be fluent in English, 2) feeling lonely (the UCLA loneliness scale, score ≥ 28), and 3) have resided in Finland for no more than five years. A total of 42 participants initially enrolled in the study; however, 3 participants withdrew before its completion. To ensure high data quality, participants who withdrew were excluded from the final analysis. The finalized dataset included 2158 observations from 39 participants. Recruitment efforts utilized purposive and snowball sampling techniques, leveraging advertisements on select social media platforms, language schools, and universities. Eligible participants were also encouraged to recommend other Finnish immigrants who might meet the inclusion criteria, further expanding the recruitment pool.

### Procedures

After confirming eligibility, participants were scheduled for an in-person meeting where they received detailed information about the study objectives and procedures, as well as instructions on the use of the provided devices. Each participant

was provided with a Samsung Watch Active 2 [22] and an OURA ring [23] to monitor cardiovascular metrics, sleep patterns, and physical activity. They were instructed to wear these devices on their non-dominant hand continuously during their daily routines, except when charging or during activities that might risk damaging the devices.

During the initial meeting, participants were also guided through the installation of two mobile applications on their smartphones: the AWARE app for tracking passive sensing data [24] and a custom-designed app – "Centralive" – for completing daily and weekly surveys [25]. Participants were required to keep their devices with them throughout the 28-day period, and they received push notifications to complete ecological momentary assessment (EMA) surveys every day. These notifications prompted participants to provide real-time data on their feelings of loneliness.

## Data collection

Two wearable devices, the Samsung Watch and the Oura Ring, were employed to gather data on cardiovascular metrics, sleep patterns, and physical activity. We programmed the Samsung Watch, equipped with Tizen open-source operating system [26], to collect raw photoplethysmogram (PPG) signals at a 20 Hz sampling frequency for 12 minutes every two hours. PPG is an optical technique that measures blood volume changes in the microvascular bed of tissue, commonly used to derive heart rate and heart rate variability. In this study, "raw" PPG refers to the unprocessed time-series waveform captured directly from the watch's optical sensor, before any filtering, reconstruction, or peak detection. The Oura Ring collected features on sleep quality, such as sleep average respiratory rate, sleep duration, and sleep efficiency, as well as physical activity features like total burned calories and inactive duration.

We utilized the AWARE application to passively capture smartphone usage and daily behavioral data. The app recorded phone interactions, location information, and screen activity patterns. To ensure participant privacy, we carefully handled sensitive location information. Exact latitude and longitude coordinates, which could reveal precise addresses, were not used in the analysis. Instead, we converted location data into generalized tags, such as "home," "work," or "other places," based on visit frequency and duration.

We also developed the Centralive app that allowed participants to complete daily and weekly surveys. The Centralive gathered passive data from the wearable devices and smartphone sensors, along with participants' responses to surveys, and uploaded it to a cloud server for secure storage.

## Measures

**Assessment of loneliness.** Loneliness as a concept has been described as having social and emotional elements, as well as encompassing intimate, relational and collective meanings [27,28]. General loneliness is most commonly measured using validated, indirect multi-item questionnaires, such as the 20-item University of California Los Angeles Loneliness Scale (UCLA) and the 11-item Rasch-Type Loneliness Scale (RTLS) [28,29]. Both instruments also have validated short forms: the 3-item UCLA scale [30] and a brief 6-item version of the RTLS [31], making them quick and practical to administer. In addition, single-item direct measures are used in research. They have been shown to be reliable. Although single-item measures are often criticized for high state variance, it is argued that, when assessing subjective constructs like loneliness, this variance may still contain substantial reliable information. Nevertheless, such measures may not fully capture the complexity of loneliness [29].

Loneliness in this study was assessed using a direct EMA questionnaire delivered five times daily. This allowed participants to report real-time loneliness states. At designated time—8 a.m., 2 p.m., 5 p.m., 8 p.m., and 10 p.m.—participants were prompted to respond to the question, "How lonely do you feel right now?" using a sliding scale from 0 (not at all) to 10 (extremely). Our single item scale was used as a measure of participants' self-reported perception of their level of loneliness, as an emotion, over time. This scale was not previously validated, however was useful for our context, sustainable for the amount of responses we required for our EMA approach, and non-ambiguous, therefore suitable for use in this

study [32]. Participants reported an average loneliness score of 3.7 (SD = 2.4). These responses provided continuous data on loneliness levels, which were subsequently used for analysis.

Despite limitations related to validity, EMA was selected over validated short multi-item scales such as UCLA 3-item scale [30] due to its capacity to capture data in real time, offering high temporal resolution through repeated assessments throughout the day. In contrast, multi-item scales rely on retrospective self-reporting and yield only a single summary score. Furthermore, EMA facilitates a higher degree of personalization by enabling the identification of individual patterns and fluctuations in loneliness, whereas multi-item scales provide a more generalized assessment.

### Derived physiological and behavioral features

We derived physiological and behavioral features from wearable devices and smartphone sensors. Initially, data collected from the wearables and smartphone sensors yields over 113 features (Table A, B, C in S1 Table in S1 File). We conducted a two-stage selection process to ensure relevance and interpretability of these features in the context of loneliness analysis.

In the first stage, we examined the causal average treatment effects (ATEs) of all features on loneliness. Features with significant causal relationships to loneliness were selected for further analysis. In the second stage, an exploratory approach was used to refine the selection. Two members of the research team—a Professor and an Assistant Professor in Nursing Science, both co-authors of this paper—reviewed the results of the CML analysis, consulting the existing literature about the relationships between loneliness, sleep, stress, physical activity and cardiac function. Features were selected based on their relevance to clinical findings from the field of loneliness screening and management.

The following subsections outline how data were processed for each source:

- **Sleep, Physical Activity, and Daily Cardiac Metrics** were monitored using the Samsung Watch and Oura Ring. Heart rate and HRV features were extracted from Samsung Watch raw PPG signals. These signals were processed through an end-to-end analysis pipeline, meaning that the raw PPG data were automatically transformed into meaningful cardiovascular features without the need for manual intervention at intermediate steps. This pipeline involves three key stages: signal quality assessment (SQA), signal reconstruction, and peak detection [33]. Initially, PPG signals were categorized into "clean" and "noisy" segments during the SQA phase [34]. Short noisy segments, those less than 15 seconds in duration, were subsequently reconstructed using a trained generative adversarial network (GAN) model [35], which included a generator and discriminator component. The generator was trained to capture the characteristic features of clean PPG signals, while the discriminator differentiated between the generated and original signals, enhancing the reconstruction accuracy. The reconstructed signals were then analyzed using a Dilated Convolution Neural Network (DCNN) to accurately detect systolic peaks and determine inter-beat intervals (IBIs) [36]. Finally, heart rate and HRV features were derived from the identified IBIs, providing a comprehensive set of cardiovascular metrics for further analysis. This end-to-end pipeline has been openly released as a Python package on PyPI [37].

  Simultaneously, the Oura Ring provided sleep and activity features. It evaluated the sleep quality by recording features such as sleep average respiratory rate and sleep efficiency. It also measured physical activity features, including energy consumption (total burned calories) and duration of being inactive.

**Smartphone Usage and Social Interactions** were captured using the AWARE application.

Smartphone usage included screen activity, including the number of times the screen was unlocked. Social interactions were derived from phone interactions, including the total duration of incoming and outgoing calls, the frequency of outgoing and missed calls, the number of received and sent messages as well as the duration spent at home.

As mentioned in the Data Collection section, due to privacy concerns, we transformed the latitude and longitude coordinates into generalized address tags using the Google Map API [38]. The "home" location was determined by identifying

the most frequent location where the participant stayed between 11 p.m. and 5 a.m. the following day. The number of distinct places visited was calculated based on these tags.

All AWARE data, including usage and social interaction features, was averaged over a 24-hour time window.

### Selected sleep features

**Average respiratory rate.** The average respiratory rate during sleep is the number of breaths per minute during sleep. The feature reflects respiratory health and autonomic nervous system function.

**Sleep efficiency.** Sleep efficiency was calculated as the ratio of total sleep time to time spent in bed, indicating the quality of sleep.

**Total sleep time.** The total sleep time is the total duration of sleep during one night, indicating an overview of the sleep quality.

### Selected physical activity features

**Total calories.** The total calories burned per day were calculated based on participants' activity levels. This feature provides an estimate of energy expenditure throughout the day, as measured by the OURA ring, which is described in detail in the subsequent subsection.

**Minutes being inactive.** This feature represents the total number of minutes per day during which participants were sedentary or engaged in low-energy activities, as also measured by the Oura Ring. Inactivity in OURA is defined as periods where the average metabolic equivalent of task (MET) level for a minute falls between 1.05 and 2, typically reflecting sitting or standing still [23].

### Selected cardiac features

- **Heart rate.** Heart rate measures were used to assess overall autonomic nervous system activity. Average sleep heart rate represents the mean heart rate during sleep periods. The average daily heart rate was calculated across all 24 hours throughout the day. It provides an overall measure of cardiac activity.

- **Time-domain HRV features.** Time-domain HRV quantifies the variations of the intervals between consecutive heartbeats over time. Average sleep root mean square of successive differences (RMSSD) was calculated as the square root of the mean of the squared differences between adjacent inter-beat intervals (IBI). This feature reflects vagal tone, which plays a critical role in emotional regulation and stress responses [39].

- **Frequency-domain HRV features.** Frequency-domain HRV features include low-frequency (LF) HRV, high-frequency (HF) HRV, and the LF/HF ratio, which were calculated using spectral analysis of heart rate intervals. HF is widely regarded as a marker of parasympathetic activity, while LF and the LF/HF ratio are often interpreted as indicators of sympathetic activity [40].

### Ethical considerations

Ethical approval for the study was obtained from the Ethics Committee for Human Sciences at the University of Turku (26/2022). Each participant provided a written informed consent before their participation in the study. For recruitment, permission for each organization was applied according to their regulations. Lonely immigrants can be considered

particularly vulnerable research subjects. Therefore, the researchers had a confidential and close contact with the participants, so that they could be referred to health services if necessary. No such need occurred during the course of the study. In addition, information on support available from the third sector was distributed to the participants. In accordance with PLOS' Best Practices in Research Reporting, we have completed the PLOS Global Research Inclusivity Questionnaire. A full copy of the completed questionnaire is provided in the Supporting Information (S1 Text in S1 File).

### Data analytic plan

Several preprocessing steps were carried out to enhance data quality. Missing data primarily resulted from participants forgetting to wear their devices or failing to charge them overnight in data collection. Contrary to the original hypothesis, Little's MCAR test indicated that the missingness in our dataset was not completely at random ($p < 0.001$). Given this, we selected the k-nearest neighbor (KNN) imputation method. This algorithm replaces missing data by averaging the values of the k closest neighbors based on feature similarity. The missing value is then estimated as the weighted average of these neighbors' values. KNN imputation allows for the flexible definition of neighbors, making it particularly suitable for our multi-modal time-series data, where local similarity is more informative than global covariance structures. Therefore, we chose a relatively small k value ($k = 3$) to avoid introducing bias, as suggested by previous study [41]. We conducted a sensitivity analysis comparing the results of KNN and Multiple Imputation (MI) methods. The feature-level ATEs and corresponding 95% confidence intervals obtained from both imputation strategies are presented in figures in S1 Fig in S1 File.

Additionally, to ensure that features across different scales were comparable, min-max scaling was applied. This scaling method transforms each feature to a specific range, typically between 0 and 1, by adjusting the values based on the feature's minimum and maximum values. To ensure proper temporal ordering and reduce the risk of reverse causality, all features were extracted from a 24-hour window preceding each EMA-based loneliness assessment.

### Causal machine learning

**Causal discovery.** We conducted causal discovery within each category of features using three methods: Peter-Clark (PC) [42], Greedy Equivalence Search (GES) [43], and Linear Non-Gaussian Acyclic Models (LINE-GAM) [44]. PC is a constraint-based algorithm that begins with a fully connected graph and removes edges by testing conditional independence among variables. This method is suitable for our dataset, given the high-dimensional features collected from wearable and smartphone sensors. GES is a score-based approach that starts with an empty graph, iteratively adding or removing edges to optimize a scoring metric like the Bayesian Information Criterion. GES is chosen for its flexibility to model both linear and non-linear relationships, which aligns with the nature of our dataset comprising behavioral and physiological features. LINE-GAM is a causal discovery method designed to identify causal relationships under the assumption of linear non-Gaussian acyclic models. This method is selected for our dataset, considering features exhibit non-Gaussian distributions under free-living monitoring.

The discovered graphs revealed no mediators or confounders within the same category of features. This finding allowed us to focus on direct causal relationships between features and loneliness. Two members of the research team (a Professor and an Assistant Professor in Nursing Science, both co-authors of this paper) reviewed the graphs and provided domain-specific insights from literature in the field. Their review helped ensure that the CML models' aligned with with existing evidence regarding the known influences of loneliness on daily physiological and behavioral features.

**Causal inference.** Based on the results of causal discovery, we specified a causal graph that modeled the hypothesized direct effects of every individual feature on loneliness. Using this graph, we estimate the ATE for each feature, treating it as the treatment variable and the loneliness score as the outcome variable. This enabled us to quantify how changes in specific features (e.g., smartphone usage, physical activity, sleep patterns) influenced participants' self-reported loneliness.

Let $Y$ represent the outcome variable (loneliness score), and $T$ represent the treatment variable (i.e., a specific feature such as heart rate, sleep duration, or number of phone unlocks). We aimed to estimate the ATE, which is defined as:

$$ATE = E[Y(T = 1)] - E[Y(T = 0)]$$

Where $E[Y(T = 1)]$ is the expected loneliness score when the treatment is applied (i.e., the feature is at a higher level), and $E[Y(T = 0)]$ is the expected loneliness score when the treatment is not applied (i.e., the feature is at a lower level).

We estimated the expected outcomes $E[Y(T = 1)]$ and $E[Y(T = 0)]$ by fitting a linear regression model using the backdoor adjustment method [45] (Sharma et al., 2021). The ATE was calculated by comparing the predicted outcomes under the treatment and control conditions:

$$ATE = \frac{1}{N} \sum_{i=1}^{N} (\hat{Y}_i (T = 1)) - \hat{Y}_i (T = 0))$$

Where $\hat{Y}_i (T = 1)$ and $\hat{Y}_i (T = 0)$ are the predicted outcomes under the treatment and control conditions for each individual.

**Refutation analysis.** We conducted refutation analysis to validate the robustness of our causal estimates. Refutation analysis involves introducing noise to the common cause variable or replacing the treatment with a random variable to challenge the estimates [46]. This analysis helps assess the robustness of the causal estimate results. If an estimator fails the refutation test (i.e., p-value < 0.05), the estimator and its assumptions are deemed invalid. We applied three refutation methods in our experiments:

Placebo Treatment Test: A random "placebo" treatment variable was generated to assess whether the ATE estimate is misleading. This test examines what happens when the true treatment variable is replaced with an independent random variable that has no causal relationship with the outcome.

Data Subset Validation: We re-estimated the ATE using randomly sampled subsets of the data to confirm the consistency of our findings across different feature subsets.

Add Random Cause: We add an independent random variable as a common cause to the dataset to confirm the robustness of the causal estimate under potential confounding.

The causal discovery, estimation, and refutation analysis were implemented by the DoWhy package [21].

# Results

## Demographic information of participants

The participants' background information is summarized in Table 1. The sample included 39 participants with an average age of 31.5 years (SD = 6.1). The gender distribution was 66.7% female (n = 26) and 33.3% male (n = 13). Participants had resided in Finland for an average of 17.7 months (SD = 15.9) and reported an average of 2.1 people around them in daily life (SD = 1.2). The baseline loneliness score, measured by EMA, averaged 3.8 (SD = 2.7).

## Causal estimates of daily-living features' impact on loneliness

Causal estimation results are presented in Table 2, with ATEs of selected daily-living features on loneliness visualized in Fig 1. Positive and negative ATEs are separately illustrated in Fig 1. Positive ATEs (left panel) indicate features that causally increase loneliness, and negative ATEs (right panel) represent features linked to decreased loneliness.

**Social Interactions** Significant positive ATEs were found for incoming call duration (ATE = 0.157, p < .001), outgoing call duration (ATE = 0.197, p < .001), outgoing call counts (ATE = 0.120, p < .001), missed call counts (ATE = 0.163, p < .001), and number of screen unlocks (ATE = 0.097, p < .001), indicating that more phone call activity causally increased

**Table 1. Participants' background information (n = 39).**

| Parameters | Values |
| --- | --- |
| Age, mean (SD) | 31.5 (6.1) |
| Gender | |
| Female | 26 |
| Male | 13 |
| Months of residence in Finland, mean (SD) | 17.7 (15.9) |
| Number of people around, mean (SD) | 2.1 (1.2) |
| Finnish language skill | |
| None | 8 |
| Basic | 26 |
| Intermediate | 5 |
| Educational attainment | |
| High School Diploma | 1 |
| Some College | 1 |
| College Degree | 11 |
| Some Graduate School | 2 |
| Graduate Degree | 24 |
| Marital status | |
| Married | 18 |
| Single | 14 |
| Cohabitating | 5 |
| Divorced | 1 |
| Prefer not to say | 1 |

loneliness. Higher number of received (ATE = 0.084, p = .001) and sent messages (ATE = 0.078, p < .001) also linked to an increase in loneliness.

Conversely, the number of places visited (ATE = −0.041, p = .009) and home duration (ATE = −0.107, p < .001) demonstrated significant negative ATEs, suggesting that either spending more time in varied places or more time at home causally reduced loneliness.

**Physical Activity** A significant negative ATE was found for more total calories burned (ATE = −0.174, p < .0001), suggesting that higher physical activity causally reduced loneliness. Conversely, more minutes of inactivity (ATE = 0.130, p < 0.001) were found to increase loneliness.

**Sleep** Two significant positive ATEs were found: average respiratory rate (ATE = 0.144, p < .001) and sleep efficiency (ATE = 0.088, p < .001), suggesting that higher respiratory rate, improved sleep efficiency and increased loneliness. One significant negative ATE was found for higher total sleep time (ATE = −0.103, p < 0.001) causally reduced loneliness.

**Cardiac features** Significant positive ATEs were observed for higher low-frequency HRV (ATE = 0.129, p < .00001), higher high-frequency HRV (ATE = 0.130, p < .00001), and higher LF/HF ratio (ATE = 0.129, p < .00001), showing a causal increase in loneliness. In contrast, average daily heart rate had a negative ATE (ATE = −0.105, p < .001), indicating that lower daily heart rates causally reduced loneliness. Cardiac features measured during sleep showed causal effects on loneliness. Higher average sleep heart rate (ATE = 0.104, p < .0001) and higher lowest sleep heart rate (ATE = 0.112, p < .001) causally increased loneliness. In contrast, average sleep RMSSD, an indicator of HRV, had a negative ATE (ATE = −0.128, p < .0001), suggesting that greater sleep RMSSD causally reduced loneliness.

**Table 2. ATEs of Features on Loneliness.**

| Category | Feature | ATE | p value | CI lower | CI upper |
|---|---|---|---|---|---|
| Social interactions | Income call durations | 0.157 | p<.001 | 0.118 | 0.203 |
| | Outgoing call duration | 0.197 | p<.001 | 0.147 | 0.248 |
| | Outgoing call counts | 0.199 | p<.001 | 0.155 | 0.252 |
| | Missed call counts | 0.163 | p<.001 | 0.119 | 0.211 |
| | Number of received messages | 0.084 | 0.001 | 0.048 | 0.126 |
| | Number of sent messages | 0.078 | p<.001 | 0.051 | 0.106 |
| | Number of places | −0.041 | 0.010 | −0.076 | −0.003 |
| | Home duration | −0.107 | p<.001 | −0.149 | −0.066 |
| | Number of screen unlocks | 0.097 | p<.001 | 0.055 | 0.135 |
| Activity | Total calories | −0.174 | p<.001 | −0.210 | −0.141 |
| | Minutes being inactive | 0.130 | p<.001 | 0.091 | 0.170 |
| Sleep | Average respiratory rate | 0.144 | p<.001 | 0.107 | 0.184 |
| | Sleep efficiency | 0.088 | p<.001 | 0.051 | 0.128 |
| | Total sleep time | −0.103 | p<.001 | −0.142 | −0.067 |
| Cardiac features | Lowest sleep heart rate | 0.112 | p<.001 | 0.077 | 0.147 |
| | Average sleep RMSSD | −0.128 | p<.001 | −0.160 | −0.096 |
| | Average sleep heart rate | 0.104 | p<.001 | 0.070 | 0.140 |
| | Low-frequency HRV | 0.129 | p<.001 | 0.092 | 0.165 |
| | High-frequency HRV | 0.130 | p<.001 | 0.091 | 0.169 |
| | LF/HF | 0.129 | p<.001 | 0.089 | 0.170 |
| | Average daily heart rate | −0.105 | p<.001 | −0.145 | −0.068 |

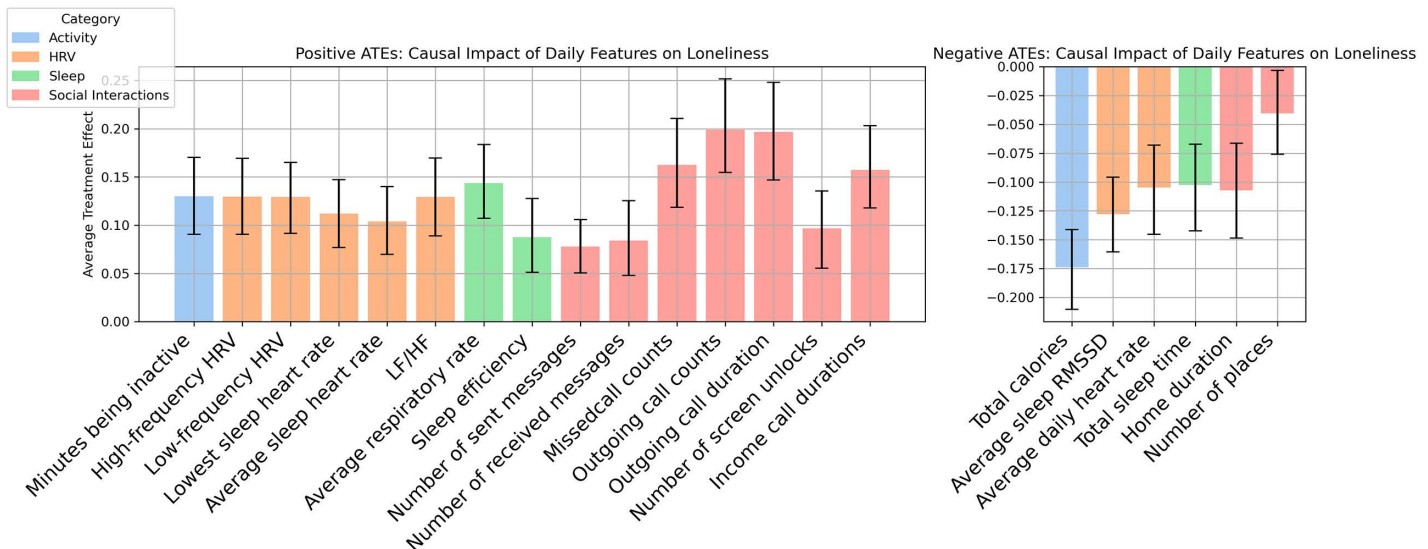

**Fig 1. ATEs of Daily Features on Loneliness Across Feature Categories.** (a) Positive ATEs, and (b) Negative ATEs.

## Discussion

This study observed important daily-living features related to loneliness for the eventual development of robust loneliness screening solutions founded on CML analysis methods with a population of first-generation immigrants in Finland. Loneliness was monitored in a sample of immigrants using a single item scale, constructed by our research team as an unambiguous measure of loneliness as an emotional state. A number of measures have been developed to monitor loneliness as it has deleterious impacts on the health of individuals [28,30,31]. Retrospective reports of the feeling of loneliness are less reliable than real-time monitoring, and up until now there have been only a limited amount of published works on the real-time screening of experiences such as loneliness, and even less implementation of this screening approach in practice. Loneliness studies commonly focus on children, young adults, and older adults, therefore knowledge about the risk factors for loneliness specific to first-generation immigrants has been lacking [8]. Our observational study shows good feasibility of using this novel approach of EMA data collection and CML analysis for the collection and interpretation of observational data related to feelings of loneliness and daily-living features.

The perceived sense of being an outsider, or socially isolated results in feelings of loneliness [8]. Individuals' relationship quality and quantity are indicators of loneliness, and can be compounded by characteristics of community support structures and patterns of socialization in living environments [8]. Individuals' vary in their preference for quantity of relationships, and level of interaction in their living environments. A screening tool using EMA and CML analysis methods shows promise for interpreting social interactions related to location and call/message activity. In our sample of first-generation immigrants in Finland we observed that increase in call activity was linked to increased feelings of loneliness. This result is likely due to the fact that there are individual differences to how much social contact is preferred by any given person. This finding underscores the importance of considering individual differences in social preferences, as some participants may prefer less frequent social contact. Furthermore, the lack of contextual data on the purpose or nature of call activity limited the ability to interpret these results fully. The number of distinct places visited and the time spent at home were both linked to reduced loneliness. These results suggest that engaging in social contact in a variety of locations and maintaining a balance between home-based activities and outings could contribute positively to well-being. This complexity supports improved specificity of measurement of social interactions, location of activities, and the investigation of immigrant perceptions of social contact/activities (i.e., phone calls, location and frequency of contact in and outside of the home). Previous literature supports the importance of considering chosen solitude as a positive mental health feature, therefore, increase in social activity/phone calls is likely not a complete metric for understanding prevention of loneliness [47]. However, it is worth noting that the estimated average treatment effect (ATE) for some features—such as Number of Places—was relatively small (i.e., −0.041), despite reaching statistical significance. This suggests that while the association may be consistent, its practical impact on loneliness is limited. Therefore, interpretations should be made cautiously, considering both the effect size and the broader behavioral context.

Our findings agree with previous research conducted to understand the relationship between smartphone use/social media app usage and loneliness. Our results reveal a link between frequency of phone unlocks and increased loneliness. Researchers have previously found that young adults' time spent on smartphones and social media apps was linked to an increase in loneliness [48]. However, other researchers in the field identified that in instances when a phone is being unlocked to connect in meaningful social ways, the relationship between loneliness and phone use might not be a maladaptive one [49]. The literature and our results inform direction for future research about the link between loneliness, use of smartphones and engagement in social media; contextual factors should be collected to support a more nuanced understanding of how, and why a phone is being unlocked [48,49].

Based on our findings, physical activity reduced loneliness. This is in line with earlier research, as benefits of physical activity have been observed in the field of health and psychology. Beyond the benefits to longevity, cardiovascular, and endocrine function physical activity has been observed to influence positive opportunities for social interaction, improved psychological health and the quality of social relationships for individuals [50,51]. Although our results agree at some level

with the current evidence about the relationship between loneliness and physical activity, the relationship between the two has been shown to be multi-factoral. Individuals that do not experience distressing effects of loneliness have higher levels of physical activity in daily life compared to lonely individuals [1,52]. Recent survey study findings from college students in the United States and older adults living in Spain align with our findings that inactivity is associated with increased loneliness [53,54]. Further, in a cross-sectional study in Sweden investigating the associations between country of birth, economic insecurity, anxiety/depression, and physical inactivity in native Swedish and Iraqi adult immigrants found that Iraqi immigrants experienced a modified negative effect of physical inactivity on their prevalence of anxiety and depression compared to their native-born counterparts [55].

Improved sleep efficiency and respiration rate during sleep were monitored daily for each participant and the influence of these factors on increased levels of loneliness was observed. Meta-analyses have been conducted to reveal medium strength associations between poor sleep and subjective and objective measures of loneliness [56,57]. The links between sleep quality, efficiency, insomnia, and nightmares and loneliness are significant [58], further increased feelings of depression related to perceptions of loss or lack of social connectedness could increase the amount of sleep individuals have each night. This could explain our finding that revealed that improved sleep efficiency increased the feelings of loneliness. The relationship between sleep duration and loneliness is unclear [56]. Our findings provide further support for the study by Christiansen and colleagues which found that sleep duration mediated the association between loneliness and morbidities such as cardiovascular diseases [59]. Based on our study, it remains unclear if the features of sleep quality as separate features can reliably be considered as a cause of loneliness. Because different features affect sleep quality in distinct ways, these features need to be linked to other features like symptoms of depression and characteristics of social interactions with more complex causal graphs.

Our findings suggest that cardiac features detected during sleep are reliable features to be linked to reduced or increased loneliness. A higher RMSSD during sleep, an indicator of maladaptive parasympathetic response to stress, was linked to decreased loneliness [39,60]. Chronic loneliness as a stressor has been associated with altered parasympathetic function, i.e., lower resting HRV and reactivity [61]. Higher sleep heart rates and sleep resting heart rates were associated with increased loneliness. These trends are also consistent with the existing study that lower sleep heart rate indicates better stress reactivity and recovery [62].

Daily cardiac features—such as daily HF-HRV, LF-HRV, LF/HF ratio, and average daily heart rate—seemed to follow consistent patterns in some respects. For example, higher HF-HRV during the day, associated with increased loneliness, aligns with the finding that higher RMSSD during sleep is linked to reduced loneliness, as both reflect parasympathetic activity. However, higher average daily heart rate causally reduces loneliness, which represents an opposite trend. These differences may be explained by the impact of daily activities—the key distinction between sleep-based and daily measurements. Daily activities, including physical movements and other psychosocial or emotional events that participants experienced throughout the day might have significantly influenced average daily heart rates and HRV. In the specific case of our immigrant participants, existing literature showed that there was an overall decrease in HF-HRV over time, but greater social integration was associated with higher HF-HRV at later follow-ups. While higher HF-HRV is typically linked to better adaptation, it may not necessarily indicate reduced loneliness in this context. For immigrants, elevated HRV could reflect better physiological adaptation to a new environment rather than an absence of loneliness [63,64]. This highlights the need to carefully interpret daily HRV data, particularly in populations undergoing significant life transitions, such as immigration.

## Implications for future research

Our study highlights the effectiveness of using objective sensing data from smartphones and wearables to uncover daily-living features associated with loneliness. For example, call activity data provided valuable insights into communication patterns, and location data revealed how diverse daily routines are linked to loneliness reduction. Cardiac metrics, such as HRV and heart rate, further enriched our understanding of the physiological correlates of loneliness. However, while these data sources were insightful, they lacked contextual details. For instance, while call activity data provided insights into

communication patterns, it lacked the contextual information for specific call activity (e.g., whether calls were domestic or international, work-related or personal). Details about the type of call that is being received or made would aid in the ability to interpret these results. Similarly, location data (whether being more at home or varied places) was linked to causally reducing loneliness, but it failed to capture the reasons for location change. Additionally, cardiac features like daily HRV and heart rate may have been influenced by personal events that were not captured in the dataset. Significant life events, interpersonal interactions, or daily stressors could have contributed to the observed results.

Our CML framework demonstrated the ability to identify direct causal relationships between individual features and loneliness. This simple structure was useful for exploring key daily-living predictors of loneliness. To support transparent interpretation, we reported 95% confidence intervals for all ATEs alongside point estimates. This enables readers to evaluate not only statistical significance but also the precision and potential real-world relevance of each effect. Some small but statistically significant ATEs may reflect consistent but subtle behavioral signals, rather than large-scale intervention targets. However, the current framework does not account for confounders across feature categories or potential mediation effects. For example, a cardiac feature might influence loneliness indirectly through its impact on sleep or social interactions, and these intermediary relationships are not captured in the current framework. Future studies can address these limitations by leveraging domain knowledge and existing research to predefine more complex and precise causal graphs. In such domain-driven Directed Acyclic Graphs (DAGs), where multiple mediators may lie between predictors and outcomes, Structural Equation Modeling (SEM) can be employed to analyze interacting or parallel mediation paths [65]. In addition, while our temporal design (i.e., proceeding 24-hour time window) mitigates reverse causality, longer-term bidirectional effects between constructs like sleep and loneliness may need further analysis in the future work.

Furthermore, the feature selection in this study was conducted by two co-authors from the research team, rather than independent experts. This may introduce bias and limit the objectivity of the results, reducing their generalizability without external validation.

Our EMA design provided frequent and consistent data on participants' emotional states, enabling timely observations of loneliness-related trends. However, the current EMA can be expanded to collect richer information about participants' behaviors. For example, incorporating event-triggered EMA at the moments of key behavior changes (e.g., when participants switch location) would provide valuable contextual data. Such questions could clarify whether the participants were engaging in social or solitary activities, whether these activities were voluntary, and how these factors affected their well-being. In addition, qualitative research could help in gaining a deeper understanding of the meanings of social interactions and loneliness in specific populations.

## Conclusion

Our novel approach for the collection and interpretation of observational data related to feelings of loneliness and daily-living features such as social interaction, activity, sleep, and cardiac features of the immigrant population showed promise to guide the development of a screening tool for risk factors of loneliness. However, these features require more contextual information to demonstrate reliable causality to increased or decreased loneliness. In the future, CML approaches should also account for potential confounders and mediators. Despite these limitations, this is an important first step towards being able to identify direct causal relationships between individual features and loneliness.

## Supporting information

**S1 File. S1 Table.** Full List of Features. Table A. List of features from Samsung watch; Table B. List of features from OURA ring; Table C. List of features extracted from smartphone. **S1 Fig.** Sensitivity Analysis. Fig A. ATE Estimates and Confidence Intervals by Feature Using KNN and MI Imputation; Fig B. Absolute ATE Difference Between KNN and Multiple Imputation. **S1 Text.** Inclusivity in Global Research Questionnaire.
(ZIP)

## Acknowledgments

We would like to extend our gratitude to Parisa Farzanehkari for her assistance in data collection for this study.

## Author contributions

**Conceptualization:** Yuning Wang.

**Data curation:** Yuning Wang, Jennifer Auxier, Mark Amayag.

**Formal analysis:** Yuning Wang.

**Methodology:** Yuning Wang, Jennifer Auxier.

**Supervision:** Iman Azimi, Amir M. Rahmani, Pasi Liljeberg, Anna Axelin.

**Validation:** Yuning Wang, Jennifer Auxier, Mark Amayag, Anna Axelin.

**Visualization:** Yuning Wang.

**Writing – original draft:** Yuning Wang, Jennifer Auxier, Anna Axelin.

**Writing – review & editing:** Mark Amayag, Iman Azimi, Amir M. Rahmani, Pasi Liljeberg, Anna Axelin.

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
