## [Decision Letter · Decision Letter 0]

30 May 2025

Dear Dr. Wang,

Thank you for submitting your manuscript to PLOS ONE. After careful consideration, we feel that it has merit but does not fully meet PLOS ONE’s publication criteria as it currently stands. Therefore, we invite you to submit a revised version of the manuscript that addresses the points raised during the review process.

Please provide a point-by-point and detailed response.

We look forward to receiving your revised manuscript.

Kind regards,

Mohammad Mofatteh, PhD, MPH, MSc, PGCert, BSc (Hons), MB BCh (c)

Academic Editor

PLOS ONE

Journal Requirements:

3. In the online submission form you indicate that your data is not available for proprietary reasons and have provided a contact point for accessing this data. Please note that your current contact point is a co-author on this manuscript. According to our Data Policy, the contact point must not be an author on the manuscript and must be an institutional contact, ideally not an individual. Please revise your data statement to a non-author institutional point of contact, such as a data access or ethics committee, and send this to us via return email. Please also include contact information for the third party organization, and please include the full citation of where the data can be found.

Additional Editor Comments:

Please provide a point-by-point and detailed response.

Reviewers' comments:

Reviewer's Responses to Questions

**Comments to the Author**

1. Is the manuscript technically sound, and do the data support the conclusions?

Reviewer #1: Yes

Reviewer #2: Yes

2. Has the statistical analysis been performed appropriately and rigorously?

Reviewer #1: Yes

Reviewer #2: Yes

3. Have the authors made all data underlying the findings in their manuscript fully available?

Reviewer #1: No

Reviewer #2: Yes

4. Is the manuscript presented in an intelligible fashion and written in standard English?

Reviewer #1: Yes

Reviewer #2: Yes

Reviewer #1: While the methods employed in this paper appear sound to me, I have some suggestions concerning the readability and comprehensibility of the manuscript:

– In the section "Data Collection", raw photoplethysmogram (PPG) signals are mentioned, but it is not explained what it represents. I kindly ask to amend this.

–In te section "Derived Physiological and Behavioral Features: Please add an explanation, what an "end-to-end analysis pipeline" is, otherwise it is unclear why its "end-to-end" nature is worth mentioning so explicitly.

– In section "Causal discovery": What exactly constitutes a nursing expert? What qualifications do they exhibit? Are they nurses? What differentiates a nurse form nursing expert? Kindly clarify.

– Also, how and why are nursing experts qualified to review graphs with a psychological variable such as loneliness as an outcome?

– In section "causal Inference", a citation is missing → it simply says "[cite]" in the textbody. Please amend and add the citation.

– In the section "Discussion", another citation has been forgotten "(; Masi et al., 2011)."

– "Discussion" section: "Increasing specificity of what is being measured with the location metric is supported by previous literature about the importance of considering when solitude is experienced as a positive event that benefits a person’s overall well-being (Weinstein et al., 2023)." This sentence is too convoluted. Additionally, its meaning does not become clear to me. What does previous literature say about what the location metric actually measures? Kindly amend this, maybe by expanding on it a little. With the information currently given, it is not quite comprehensible.

– "Discussion section: You write "Immigrants specifically struggle with timely convergence with native-born peers". This sentence sounds very strange. Do you mean that immigrants struggle to come into contact or integrate with non-immigrants? Please edit this sentence to convey the intended meaning more clearly.

– "Discussion" section: You write "Further, in a cross-sectional study in Sweden investigating the associations between country of birth, economic insecurity, anxiety/depression, and physical inactivity in native Swedish and Iraqi

adult immigrants an association between physical inactivity and anxiety and depression (Siddiqui et al., 2014), both mental conditions that have been seen to influence in social experiences and perceptions of social isolation for immigrants." This sentence is too long and too convoluted. After reading it repeatedly, I'm still not sure it makes sense. Please break it down into more digestible smaller sentences that convey your intended meaning more clearly.

– "Discussion" section: I have the same issue with this sentence: "A higher RMSSD during sleep was linked to

decreased loneliness that is logical as RMSSD is a feature that tells us how well someone is coping with the parasympathetic responses to stress".

A final remark concerning data availability: If your data cannot be shared, at least make them findable in a public repository. Previous research has shown that data and code "available on request" are usually not available anymore, with researchers switching institutions an email addresses, losing access to the data themselves, do not react to these request for some reason, etc. Even if the data cannot be made public, they should follow FAIR principles and be findable at least.

Reviewer #2: Identifying Daily-Living Features Related to Loneliness: a Causal Machine Learning Approach

The authors have inferred that daily-living features, including social interactions, activity, sleep, and cardiac features, causally influence loneliness. Overall, their findings provide a basis for loneliness screening targeting immigrant populations. While I appreciate the authors for this interesting study, I have some suggestions as given below:

• While the authors found that single-item ecological momentary assessment (EMA) is feasible, what merits does it hold over validated multi-item scales like De Jong Gierveld Loneliness Scale (6-item) or UCLA-3 item scale need to be discussed in terms of the challenges and trade-offs associated with these scales.

• To make the findings robust, it’s essential to first verify the assumption of random missingness using Little’s MCAR test. The authors can compare KNN imputation with methods like multiple imputation or expectation-maximization to assess the robustness of results. Including a brief sensitivity analysis across these approaches would improve transparency and confidence in the study’s conclusions.

• I appreciate the authors for accepting that the “the current framework does not account for confounders across feature categories or potential mediation effects.”. However, it would be beneficial for the future researchers if suitable models (e.g., domain-driven Directed Acyclic Graphs or structural equation modeling) are suggested to determine the inter-category mediators.

• The authors could explain the reverse causality or bidirectional effects between sleep and loneliness. If possible, a temporal lag analysis can be conducted.

• The paper reports average treatment effects (ATEs) solely in terms of point estimates and p-values, without providing any confidence intervals. This omission is problematic, as p-values alone do not convey the precision or reliability of an effect estimate.

• Some language improvements are suggested: (i)Terms like “non-ambiguous” can distract readers and reduce the perceived professionalism of the work. It should be “unambiguous”. (ii) “The amount of relationships and the quality of relationships…” Avoid redundancy. "The number and quality of relationships..." sounds better. (iii) Minor Spacing and Typographical Issues are there. Example “… was lacking (; Masi et al., 2011).”

**Do you want your identity to be public for this peer review?** For information about this choice, including consent withdrawal, please see our Privacy Policy

Reviewer #1: No

Reviewer #2: No

---

## [Author Response · Author response to Decision Letter 1]

11 Jul 2025

Dear Dr. Mofatteh and Reviewers,

We thank you for the opportunity to revise our manuscript “Identifying Daily-Living Features Related to Loneliness: a Causal Machine Learning Approach” (PONE-D-25-06716), for consideration for publication in PLOS One. We are delighted to receive feedback from the Reviewer, as well as to address their remaining comments. In this letter we provide a point-by-point response to their comments.

Reviewer #1’s comments:

1. While the methods employed in this paper appear sound to me, I have some suggestions concerning the readability and comprehensibility of the manuscript:

In the section "Data Collection", raw photoplethysmogram (PPG) signals are mentioned, but it is not explained what it represents. I kindly ask to amend this.

Thank you for this comment. We have added a brief explanation of what “raw PPG signals” represents. We explain that PPG is an optical technique for measuring blood volume changes in tissue, and that in our study, “raw PPG” refers to the unprocessed waveform data collected directly from the Samsung Watch’s optical sensor, prior to any filtering or analysis.

2. In the section "Derived Physiological and Behavioral Features: Please add an explanation, what an "end-to-end analysis pipeline" is, otherwise it is unclear why its "end-to-end" nature is worth mentioning so explicitly.

Thank you for this comment. We have clarified that “end-to-end analysis pipeline” refers to the automated transformation of raw PPG signals into cardiovascular features. We have also included a reference to the public release of our pipeline as a Python package on PyPI.

3. In section "Causal discovery": What exactly constitutes a nursing expert? What qualifications do they exhibit? Are they nurses? What differentiates a nurse form nursing expert? Kindly clarify.

Also, how and why are nursing experts qualified to review graphs with a psychological variable such as loneliness as an outcome?

Thank you for this comment, it is apparent that our description of roles in the ‘Causal discovery’ section was not clear. The nurse experts (N=2) we referred to are members of the research team (Professor and Assistant Professor in Nursing Science) and confirmed domain-specific insights based on literature in the field and expertise in health science research. This section has been revised to aid in clarity and as ‘nursing experts’ was a vague descriptor we have removed this. For transparency, we have already included the full list of initial features considered in the first stage in Appendix Tables 3, 4, and 5, as noted in the Methods section. This allows readers to view all variables prior to domain-informed refinement and supports reproducibility.

4. In section "causal Inference", a citation is missing → it simply says "[cite]" in the textbody. Please amend and add the citation.

Thanks for pointing this out. We have added the citation in the reference.

5. In the section "Discussion", another citation has been forgotten "(; Masi et al., 2011)."

Thank you very much for pointing this out, we regret that this was a typo, sorry for the lack of clarity. We have removed the typo.

6. "Discussion" section: "Increasing specificity of what is being measured with the location metric is supported by previous literature about the importance of considering when solitude is experienced as a positive event that benefits a person’s overall well-being (Weinstein et al., 2023)." This sentence is too convoluted. Additionally, its meaning does not become clear to me. What does previous literature say about what the location metric actually measures? Kindly amend this, maybe by expanding on it a little. With the information currently given, it is not quite comprehensible.

Thank you for this comment, the writing structure has been adjusted, and ideas in the paragraph have been carefully revised to improve the argument building in this paragraph. This section of the discussion is more clear after revision and we are glad to have had the opportunity to refine the description of our interpretation of social activity (phone calls, in and out of the house, and the link to chosen solitude). We trust this will be clearer and appreciate the suggestion!

7. "Discussion section: You write "Immigrants specifically struggle with timely convergence with native-born peers". This sentence sounds very strange. Do you mean that immigrants struggle to come into contact or integrate with non-immigrants? Please edit this sentence to convey the intended meaning more clearly.

Thank you for this comment, the attention to our writing clarity with the examples have been critical for the revision and refinement of our discussion arguments. We assess our entire discussion section as being greatly improved as a result of your feedback. This paragraph has been revised further for clarity and stronger arguments, the final sentence was unclear, we agree. We have changed this and hope that we have addressed your point to your satisfaction.

8. "Discussion" section: You write "Further, in a cross-sectional study in Sweden investigating the associations between country of birth, economic insecurity, anxiety/depression, and physical inactivity in native Swedish and Iraqi adult immigrants an association between physical inactivity and anxiety and depression (Siddiqui et al., 2014), both mental conditions that have been seen to influence in social experiences and perceptions of social isolation for immigrants." This sentence is too long and too convoluted. After reading it repeatedly, I'm still not sure it makes sense. Please break it down into more digestible smaller sentences that convey your intended meaning more clearly.

(See response to comment #7): “Thank you for this comment, the attention to our writing clarity with the examples have been critical for the revision and refinement of our discussion arguments. We assess our entire discussion section as being greatly improved as a result of your feedback. This paragraph has been revised further for clarity and stronger arguments, the final sentence was unclear, we agree. We have changed this and hope that we have addressed your point to your satisfaction.”

9. "Discussion" section: I have the same issue with this sentence: "A higher RMSSD during sleep was linked to decreased loneliness that is logical as RMSSD is a feature that tells us how well someone is coping with the parasympathetic responses to stress".

Thank you for this comment. We agree and have revised the writing style in this paragraph and others throughout the discussion to improve the state of readability of the content in the discussion. We are very grateful for your comments on the long and unclear sentence structure.

10. A final remark concerning data availability: If your data cannot be shared, at least make them findable in a public repository. Previous research has shown that data and code "available on request" are usually not available anymore, with researchers switching institutions an email addresses, losing access to the data themselves, do not react to these request for some reason, etc. Even if the data cannot be made public, they should follow FAIR principles and be findable at least.

We fully agree with the importance of following the FAIR principles and ensuring long-term accessibility of research data. At the time of submission, our dataset was undergoing review by the Dryad repository. Now we are happy to inform that the review process has since been completed, and the dataset is now fully accessible online via the following DOI: https://doi.org/10.5061/dryad.qz612jmrn

Reviewer #2’s comments

The authors have inferred that daily-living features, including social interactions, activity, sleep, and cardiac features, causally influence loneliness. Overall, their findings provide a basis for loneliness screening targeting immigrant populations. While I appreciate the authors for this interesting study, I have some suggestions as given below:

1. While the authors found that single-item ecological momentary assessment (EMA) is feasible, what merits does it hold over validated multi-item scales like De Jong Gierveld Loneliness Scale (6-item) or UCLA-3 item scale need to be discussed in terms of the challenges and trade-offs associated with these scales.

We thank the reviewer for this important question.

Ecological Momentary Assessment (EMA) and validated multi-item scales, such as the UCLA 3-Item Scale and the De Jong Gierveld 6-Item Loneliness Scale, differ in their approaches to measuring loneliness. EMA captures data in real-time or near real-time, offering high temporal resolution through multiple assessments per day, whereas multi-item scales rely on retrospective self-reporting and provide only a single score. EMA enhances ecological validity by assessing experiences in natural environments, while multi-item scales lack this contextual depth and typically offer only moderate ecological validity. Additionally, EMA supports high personalization by identifying individual patterns and fluctuations in loneliness, whereas multi-item scales are more generalized. However, EMA typically requires technology, such as a smartphone app, making it more complex to implement, while multi-item scales are simpler and can be administered using paper or basic digital forms.

We have clarified our choice of EMA in the Materials and Methods/Measures/Assessment of Loneliness section.

“Despite limitations related to validity, EMA was selected over validated short multi-item scales such as UCLA 3-item scale (Hughes et al. 2004) due to its capacity to capture data in real time, offering high temporal resolution through repeated assessments throughout the day. In contrast, multi-item scales rely on retrospective self-reporting and yield only a single summary score. Furthermore, EMA facilitates a higher degree of personalization by enabling the identification of individual patterns and fluctuations in loneliness, whereas multi-item scales provide a more generalized assessment.”

2. To make the findings robust, it’s essential to first verify the assumption of random missingness using Little’s MCAR test. The authors can compare KNN imputation with methods like multiple imputation or expectation-maximization to assess the robustness of results. Including a brief sensitivity analysis across these approaches would improve transparency and confidence in the study’s conclusions.

Thank you for pointing out this important question. After we performed Little’s MCAR to our dataset, the result indicated that the missingness in our dataset was not MCAR (p < 0.001). We have addressed the concerns in the revised manuscript. Specifically:

● We applied Little’s MCAR test and found that the missing data in our dataset were not missing completely at random.

● Considering the data distribution and to minimize imputation bias, we used the KNN method and reduced the k value to a relatively small k value (from k = 5 to k = 3), as recommended in the literature. This resulted in small numerical changes to several ATE estimates, but all effect directions and statistical conclusions remain unchanged.

● We did not use Expectation-Maximization (EM) because many features in our dataset were not normally distributed and showed multiple peaks, which violates EM’s assumptions.

● While Multiple Imputation (MI) is widely used, it is best suited for data where the missingness structure can be effectively modeled with global relationships. In contrast, our data contain multi-modal distributions and time series characteristics. MI does not easily allow the direct modeling of local dependencies in such data.

● To ensure the robustness of our findings, we performed a sensitivity analysis by comparing the KNN approach with MI. Results for both methods, including the ATEs and confidence intervals, are presented in Appendix Figure S1 and S2.

We have updated the Materials and Methods/Data Analytic Plan section to clarify the imputation strategy and included all relevant justifications.

3. I appreciate the authors for accepting that the “the current framework does not account for confounders across feature categories or potential mediation effects.”. However, it would be beneficial for the future researchers if suitable models (e.g., domain-driven Directed Acyclic Graphs or structural equation modeling) are suggested to determine the inter-category mediators.

Thank you for this comment. We have expanded the “Implications for Future Research” section and recommend the future methodological direction.

4. The authors could explain the reverse causality or bidirectional effects between sleep and loneliness. If possible, a temporal lag analysis can be conducted.

Thanks for the comment. We agree that sleep and loneliness may exhibit bidirectional relationships over longer timescales. However, in our current analysis, we specifically designed the feature extraction window to address this concern. All behavioral and physiological features were aggregated over the 24-hour period preceding each loneliness assessment, which ensures a temporal ordering from treatment (every feature) to outcome (loneliness). As such, reverse causality (i.e., loneliness influencing sleep) is not applicable within the scope of this analysis, since the loneliness score was measured after all features were recorded. We have added the text to clarify this point in the Materials and Methods/Data Analytic Plan section, and discussed briefly in the Discussion/Implication for future work section.

5. The paper reports average treatment effects (ATEs) solely in terms of point estimates and p-values, without providing any confidence intervals. This omission is problematic, as p-values alone do not convey the precision or reliability of an effect estimate.

Thank you for the comment. We have added the 95% confidence intervals to Table 2. ATEs of Features on Loneliness and corresponding error bars to Figure 1. ATEs of Daily Features on Loneliness Across Feature Categories: (a) Positive ATEs, and (b) Negative ATEs.

6. Some language improvements are suggested: (i)Terms like “non-ambiguous” can distract readers and reduce the perceived professionalism of the work. It should be “unambiguous”. (ii) “The amount of relationships and the quality of relationships…” Avoid redundancy. "The number and quality of relationships..." sounds better. (iii) Minor Spacing and Typographical Issues are there. Example “… was lacking (; Masi et al., 2011).”

Thank you for this comment, we appreciate this observation and have taken a careful look throughout the manuscript and revised the examples you mentioned and cleared up any others we found along the way. I feel this has improved the manuscript’s readability and professional tone.

---

## [Decision Letter · Decision Letter 1]

6 Aug 2025

Dear Dr. Wang,

Thank you for submitting your manuscript to PLOS ONE. After careful consideration, we feel that it has merit but does not fully meet PLOS ONE’s publication criteria as it currently stands. Therefore, we invite you to submit a revised version of the manuscript that addresses the points raised during the review process.

*
**Please provide a point-by-point response to additional comments.**
*

We look forward to receiving your revised manuscript.

Kind regards,

Mohammad Mofatteh, PhD, MPH, MSc, PGCert, BSc (Hons), MB BCh (c)

Academic Editor

PLOS ONE

Journal Requirements:

Additional Editor Comments:

Please provide a point-by-point response to additional comments.

Reviewers' comments:

Reviewer's Responses to Questions

**Comments to the Author**

Reviewer #1: (No Response)

2. Is the manuscript technically sound, and do the data support the conclusions?

Reviewer #1: Yes

3. Has the statistical analysis been performed appropriately and rigorously?

Reviewer #1: Yes

4. Have the authors made all data underlying the findings in their manuscript fully available?

Reviewer #1: Yes

5. Is the manuscript presented in an intelligible fashion and written in standard English?

Reviewer #1: Yes

Reviewer #1: Dear authors,

I would like to thank you for revising your manuscript in such a thoughtful manner. I particularly appreciate your clear explanations on PPG signals and the overall improvements in the writing flow! Also, great news on the data availability – thank you and well done.

Two issues remain open for me.

1. The issue of the nursing expert / health science researchers remains a little unresolved to me. The term "nursing experts" has been changed to "health science researchers" (except in section "Derived Physiological and Behavioural Features").

But just changing the label for the experts does not fully resolve the issue: First of all, health science researchers is just as vague as nursing experts. I think the more important detail to disclose is that the two experts are not independent experts, but members of your team. This should be addressed in the paper for transparency.

1.1 Are they also co-authors of this paper? If so, that should be disclosed in the paper as well.

1.2 In the response to the review you also disclosed the number of experts (n = 2), but in the paper state "a team of" [...] experts. It seems a bit misleading to refer to two people as a team, as it makes it sound as if you had an entire team of independent experts evaluate your work.

Please note that I do not wish to torture you with these seemingly minor concerns. As researchers, we are often constrained by limited resources and have to rely on the help and support of close colleagues and team members. This is not an issue in itself, but should be disclosed honestly and transparently.

2. One methodological note: The interpretation of the ATEs in Table 2 appears to rely heavily on statistical significance (p-values), even in cases where the effect size is close to zero. For example, the ATE for Number of Places is -0.041. While statistically significant, it can be questioned whether the effect is practically relevant at all.

Yet, the authors put effort into interpreting the contradiction of this significant ATE with the somewhat contradictory significant negative effect of Home Duration.

In general, the sizes of the effects were not addressed anywhere in the paper, if I recall correctly.

**Do you want your identity to be public for this peer review?** For information about this choice, including consent withdrawal, please see our Privacy Policy

Reviewer #1: No

---

## [Author Response · Author response to Decision Letter 2]

12 Aug 2025

Dear Editor and Reviewer,

We thank you for the opportunity to revise our manuscript “Identifying Daily-Living Features Related to Loneliness: a Causal Machine Learning Approach” (PONE-D-25-06716), for consideration for publication in PLOS One. We are delighted to receive feedback from the Reviewer, as well as to address their remaining comments. In this letter we provide a point-by-point response to their comments.

Reviewer #1: Dear authors,

I would like to thank you for revising your manuscript in such a thoughtful manner. I particularly appreciate your clear explanations on PPG signals and the overall improvements in the writing flow! Also, great news on the data availability – thank you and well done.

Two issues remain open for me.

1. The issue of the nursing expert / health science researchers remains a little unresolved to me. The term "nursing experts" has been changed to "health science researchers" (except in section "Derived Physiological and Behavioural Features").

But just changing the label for the experts does not fully resolve the issue: First of all, health science researchers is just as vague as nursing experts. I think the more important detail to disclose is that the two experts are not independent experts, but members of your team. This should be addressed in the paper for transparency.

1.1 Are they also co-authors of this paper? If so, that should be disclosed in the paper as well.

1.2 In the response to the review you also disclosed the number of experts (n = 2), but in the paper state "a team of" [...] experts. It seems a bit misleading to refer to two people as a team, as it makes it sound as if you had an entire team of independent experts evaluate your work.

Please note that I do not wish to torture you with these seemingly minor concerns. As researchers, we are often constrained by limited resources and have to rely on the help and support of close colleagues and team members. This is not an issue in itself, but should be disclosed honestly and transparently.

- Thank you for these comments and your kind clarification! We fully agree and appreciate your suggestion regarding the transparency. In the revised manuscript, we have addressed the concerns as follows:

● We now clearly state that the two individuals who reviewed the causal graphs and contributed domain-specific insight are both co-authors of the paper and members of the research team (a Professor and an Assistant Professor in Nursing Science).

● We have removed vague terms such as “nursing experts” and “a team of experts” throughout the manuscript and replaced them with a clear description of their roles. Specifically:

○ In the Materials and Methods/Measures/Derived Physiological and Behavioral Features/paragraph 2 and Data Analytic Plan/Causal machine learning/Causal discovery/paragraph 2, we revised the text to state: “Two members of the research team (a Professor and an Assistant Professor in Nursing Science, both co-authors of this paper) reviewed the…”

● In the Discussion/Implications for future research, we added the following statement: “Furthermore, the feature selection in this study was conducted by two co-authors from the research team, rather than independent experts. This may introduce bias and limit the objectivity of the results, reducing their generalizability without external validation.”

2. One methodological note: The interpretation of the ATEs in Table 2 appears to rely heavily on statistical significance (p-values), even in cases where the effect size is close to zero. For example, the ATE for Number of Places is -0.041. While statistically significant, it can be questioned whether the effect is practically relevant at all.

Yet, the authors put effort into interpreting the contradiction of this significant ATE with the somewhat contradictory significant negative effect of Home Duration.

In general, the sizes of the effects were not addressed anywhere in the paper, if I recall correctly.

- Thank you for your observation. We agree that statistical significance alone does not guarantee practical or clinical relevance, and that small effect sizes (e.g., the ATE for Number of Places (-0.041)) should be interpreted with caution. We addressed the concerns in the revised manuscript. Specifically:

● We have added 95% confidence intervals for all ATE estimates in Table 2, and included error bars in Figure 1. These additions provide readers with clearer insight into the precision and variability of our estimates. The confidence intervals can help the readers judge the robustness and potential practical relevance of each finding beyond only p-values.

● In the Discussion/paragraph 2, where we discussed the Home Duration and Number of Places, we added the following statement: “However, it is worth noting that the estimated average treatment effect (ATE) for some features—such as Number of Places—was relatively small (i.e., -0.041), despite reaching statistical significance. This suggests that while the association may be consistent, its practical impact on loneliness is limited. Therefore, interpretations should be made cautiously, considering both the effect size and the broader behavioral context.”

● In the Discussion/Implications for future research/paragraph 2, we emphasized: “Some small but statistically significant ATEs may reflect consistent but subtle behavioral signals, rather than large-scale intervention targets.”

---

## [Decision Letter · Decision Letter 2]

16 Nov 2025

Identifying Daily-Living Features Related to Loneliness: a Causal Machine Learning Approach

PONE-D-25-06716R2

Dear Dr. Wang,

We’re pleased to inform you that your manuscript has been judged scientifically suitable for publication and will be formally accepted for publication once it meets all outstanding technical requirements.

Kind regards,

Laura Kelly, PhD

Division Editor

PLOS One

Additional Editor Comments (optional):

Reviewers' comments:

Reviewer's Responses to Questions

**Comments to the Author**

Reviewer #1: All comments have been addressed

2. Is the manuscript technically sound, and do the data support the conclusions?

Reviewer #1: Yes

3. Has the statistical analysis been performed appropriately and rigorously?

Reviewer #1: Yes

4. Have the authors made all data underlying the findings in their manuscript fully available?

Reviewer #1: Yes

5. Is the manuscript presented in an intelligible fashion and written in standard English?

Reviewer #1: Yes

Reviewer #1: Dear authors,

I want to thank you for your patience. All my concerns have been addressed and cleared up. In my opinion, this paper is ready for publication.

**Do you want your identity to be public for this peer review?** For information about this choice, including consent withdrawal, please see our Privacy Policy

Reviewer #1: No

---

## [Editor Report · Acceptance letter]

PONE-D-25-06716R2

PLOS One

Dear Dr. Wang,

I'm pleased to inform you that your manuscript has been deemed suitable for publication in PLOS One. Congratulations! Your manuscript is now being handed over to our production team.

Kind regards,

on behalf of

Dr. Laura Hannah Kelly

Staff Editor

PLOS One